# Prototype of running clinical trials in an untrustworthy environment using blockchain

Daniel R. Wong[1,2], Sanchita Bhattacharya[1,2] & Atul J. Butte [1,2,3]

Monitoring and ensuring the integrity of data within the clinical trial process is currently not always feasible with the current research system. We propose a blockchain-based system to make data collected in the clinical trial process immutable, traceable, and potentially more trustworthy. We use raw data from a real completed clinical trial, simulate the trial onto a proof of concept web portal service, and test its resilience to data tampering. We also assess its prospects to provide a traceable and useful audit trail of trial data for regulators, and a flexible service for all members within the clinical trials network. We also improve the way adverse events are currently reported. In conclusion, we advocate that this service could offer an improvement in clinical trial data management, and could bolster trust in the clinical research process and the ease at which regulators can oversee trials.

---

[1] Bakar Computational Health Sciences Institute, University of California, San Francisco, CA 94158, USA. [2] Department of Pediatrics, University of California, San Francisco, CA 94158, USA. [3] Center for Data-Driven Insights and Innovation, University of California, Office of the President, Oakland, CA 94607, USA. Correspondence and requests for materials should be addressed to A.J.B. (email: atul.butte@ucsf.edu)

Clinical trial networks often involve many parties and sites, and a large flow of information and confidential data. With the involvement of more parties, more exchanges, and trials being conducted far from sponsoring institutions comes the opportunity for human-induced error, whether it is unintentional or malicious. In a survey of authors of clinical drug trials, 17% of them reported that they were personally aware of intentional fabrication in research[1]. Although it is obviously not possible to calculate the actual rate or impact of fraud, there is evidence that some misconduct and alteration of data is happening in scientific and medical research[2], potentially including clinical trials. In addition, current clinical trials are executed with many manual processes that could be prone to error. Enforcing Good Clinical Practice guidelines set forth by the International Council for Harmonization of Requirements for Pharmaceuticals for Human Use (ICH) has been a challenge for regulators of trials, and process improvement is an active area of research[3]. For regulators of trials such as the Food and Drug Administration (FDA), auditing of data are challenging, and real-time oversight of a trial is lacking as there is no easy and secure way of accessing or viewing the complex network of data transactions as they occur. There can be a lack of transparency and traceability of data, lack of real-time access to results as they are made, and potentially a risk of data tampering[3]. The ability to easily trace data back to the original source can be limited[4], and the FDA has identified this lack of traceability as one of their top data issues to address[5].

Blockchain is a new software development methodology involving a unique data structure that has garnered increased attention due to the seminal paper that outlined Bitcoin (bitcoin.org)[6]. The technology provides a data structure that ensure a secure and unfalsifiable transaction history. This is accomplished primarily through the use of cryptographic hashing, which has properties and use cases in many domains ranging from internet security to banking. A hash function is a function that maps data of variable length data to a fixed-length digest. Any change to the input data result in an unpredictable change in the hash. In this implementation, each new block added to the chain includes a hash of the previous block. If the previous block is later changed, the subsequent hash would no longer be valid. In addition, blockchains are designed to be append only, and are thus immutable by design, providing a guarantee of safeguarded data. This yields a verifiable and tamper proof history of all transactions since its beginning[6].

We propose a solution to the challenges in the current clinical trials system by using blockchain technology, coupled with changes in methodology for the management of clinical trials. Due to the innate need of clinical trials to have a centralized authority, such as through the regulator, a completely decentralized blockchain infrastructure as used in Bitcoin might not be appropriate. Instead, we borrow ideas and repurpose the methodology to work with clinical trial management, and demonstrate the feasibility with a new prototype web portal. Blockchain has already been proposed for use in various healthcare settings[7], with potential applications in medical record management, claims processing, health supply chain management, and integration of geospatial data in various data modalities[8]. Work has already begun in seeing how clinical trial management can be improved with blockchain technologies[9,10]. Moreover, certain aspects relevant to the clinical trial process, such as patient recruiting and IRB (Institutional Review Board) enforcement of human subject regulations via smart contracts, have been described at a high level in the literature[11]. We improve upon previous work by implementing a web-based portal accessible to all parties with a real clinical trial dataset, facilitating and verifying patient and clinical investigator interaction, integrating version control into the blockchain, expediting adverse event reporting, and testing malicious attacks to data integrity with real world clinical data.

## Results

**Network protocol**. We modeled a prototype phase II clinical trial. Upon approval of the study protocol and initiation of phase II, we propose a future regulator could instantiate a private blockchain and registers all participating parties in the portal providing authenticated and controlled web-based and API (application programming interface) access to the blockchain. All parties would be required to use the portal service for any and all exchange of information related to the trial, and only the information present on the blockchain would be used for review when considering approval of the treatment. The network and representative transactions are illustrated in Fig. 1.

For new patient recruitment and consent acquisition by the clinical site, we propose that an Interactive Voice Response System (IVRS) generates unique verification codes for each subject to give to the trial investigator at the clinical site, and posts encrypted decoding keys for later unblinding. The decoding keys describe the various treatment types that a patient can receive, and will be saved in a password protected environment by the IVRS service provider. The trial sponsor then sends a blinded treatment distribution scheme to the trial investigator at the clinical site. The unique verification codes for each subject are appended to that subject's CRF at the clinical site upon the office visit. All CRFs would be completed digitally and considered valid if the proper verification code is present. Once completed, the CRF will be directed to the Clinical Research Organization (CRO) involved in the trial, and this transaction will be stamped onto the growing blockchain.

When adverse events are reported, we envision the Data Safety Management Board (DSMB) or regulatory agency would be able to continuously see these events through each chain-posted CRF, and then potentially propagate these onto that trial's page for public view, if appropriate. Once the CRO receives raw CRF data, data cleaning, and statistical analysis can begin and be done transparently. Upon completion, cleaned data and analysis scripts are sent to the trial sponsor through the portal and these transactions are subsequently recorded onto the blockchain. Any outside data collection sources that are enrolled in the clinical trial would also have to send data to the sponsor through the portal, all marked on the growing ledger.

When the trial sponsor wishes to apply for approval of the drug, the sponsor would send all of their finalized data and in house statistical analysis results to the regulator through the portal and these, like all other elements, would be subsequently added to the chain. When reviewing for approval, the regulator will only consider data that are present on this secure blockchain, and has full read access to everything that has occurred since the blockchain's instantiation. All data that were ever transmitted in the network would be easily accessible, and its integrity and guarantee of when the transaction occurred will be assured.

**Data transaction details**. Whenever a transaction occurs, the sender, receiver, timestamp, file attachment, and hash of the previous block, are all recorded onto a new block. These elements are then concatenated together, and hashed using the SHA256 algorithm[12], with the result instantiated as the hash string of the current block. The blockchain is constructed by creating a linked list of such blocks (Fig. 2). The previous block's hash is kept for ordering and to make each block dependent on all blocks that preceded it in the chain, which is a useful property for quickly validating a chain[6]. Data storage of the blockchain will be accomplished by duplicating and distributing the chain to

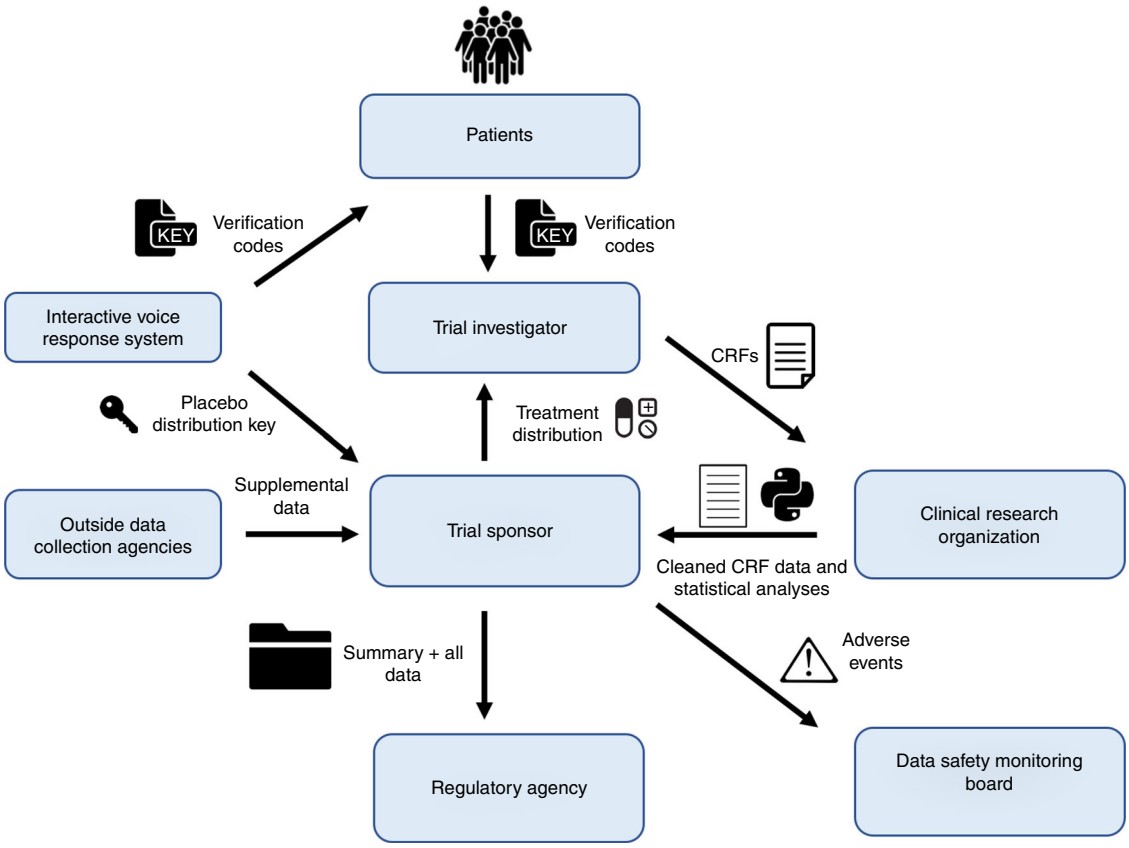

**Fig. 1** The idealized clinical trial network in the context of a blockchain-based record system. The various transactions (along each arrow) and key participants (boxed) within a clinical trial are shown

physically separate machines and data warehouses to be managed by the regulator (see Supplementary Methods).

Encryption through a password based key derivation function is offered, and can ensure that sensitive information is protected if the user chooses to do so, which is especially relevant to maintaining integrity of health and medical information, and eliminating information exposure to unwanted parties. Data are thus stored as an unintelligible series of bytes at the storage level, ensuring that any sensitive information in the network is obfuscated and will not be compromised in the incident of a data breach.

File storage of any type onto the blockchain is supported, and the user is able to encrypt, send, and extract files easily. For regulators, the full transaction history since the block's genesis is readily available with precise timestamps (Fig. 3a), and the auditing process can be done swiftly and with the confidence that all data are original or version controlled. Content since the earliest phases of the trial are sorted, fully transparent, and easily compressible, and downloadable.

**Version controls**. If a user needs to edit content that is already present on the blockchain, such as the case when an honest mistake is made and needs to be corrected, the user could make an update known by submitting a new transaction with the corrected data without overwriting the old data. By nature, blockchains are append only, so editing the data directly on the blockchain is not possible. We propose combining blockchain's append only criteria with version controlling similar to the functionality of GitHub to accommodate this issue. When a new file is uploaded in a transaction, its contents are hashed and compared to any existing files on the blockchain. If there is a conflict, then the system initiates a schema in which subsequent

and differing versions of a file are given incrementing version numbers automatically. Hence, a user can be assured that any downstream modifications to that user's file by anyone else in the network will be documented and cannot be done discretely. No trust in any other parties in the network is needed for data purity, as any tampering will be version controlled and any editors of the file will be easily identified (Fig. 3b).

**Simulation of a previously completed clinical trial**. To test how blockchain software technologies could be used to manage the governance and data management aspects of a clinical trial, we simulated how a previously completed clinical trial testing the efficacy and safety of omalizumab[13] could have been executed using blockchain. We downloaded the completed clinical trials data, including all necessary components, such as raw data, case report form (CRF) components, and protocols from the open clinical trials data repository ImmPort[14] (see Data Availability). The trial simulation sequence of events and corresponding files are shown in Fig. 2. Of the 159 actual patients in the trial with CRF data, we mimicked one subject from each of the four treatment arms for the sake of clarity. Only a few selected categories from the large wealth of CRF information were mirrored for the same reason (see Methods). The statistical scripts in this simulation are not the real Python analyses because of our lack of access.

Here, we show a simulation of how different types of clinical trial events were implemented using our blockchain-based data portal. The first event occurs during encounters between a clinical investigator and the patients after being enrolled for a clinical trial. The second event we simulated is the mutation of CRF data by the trial sponsor. The third event is a storage level corruption

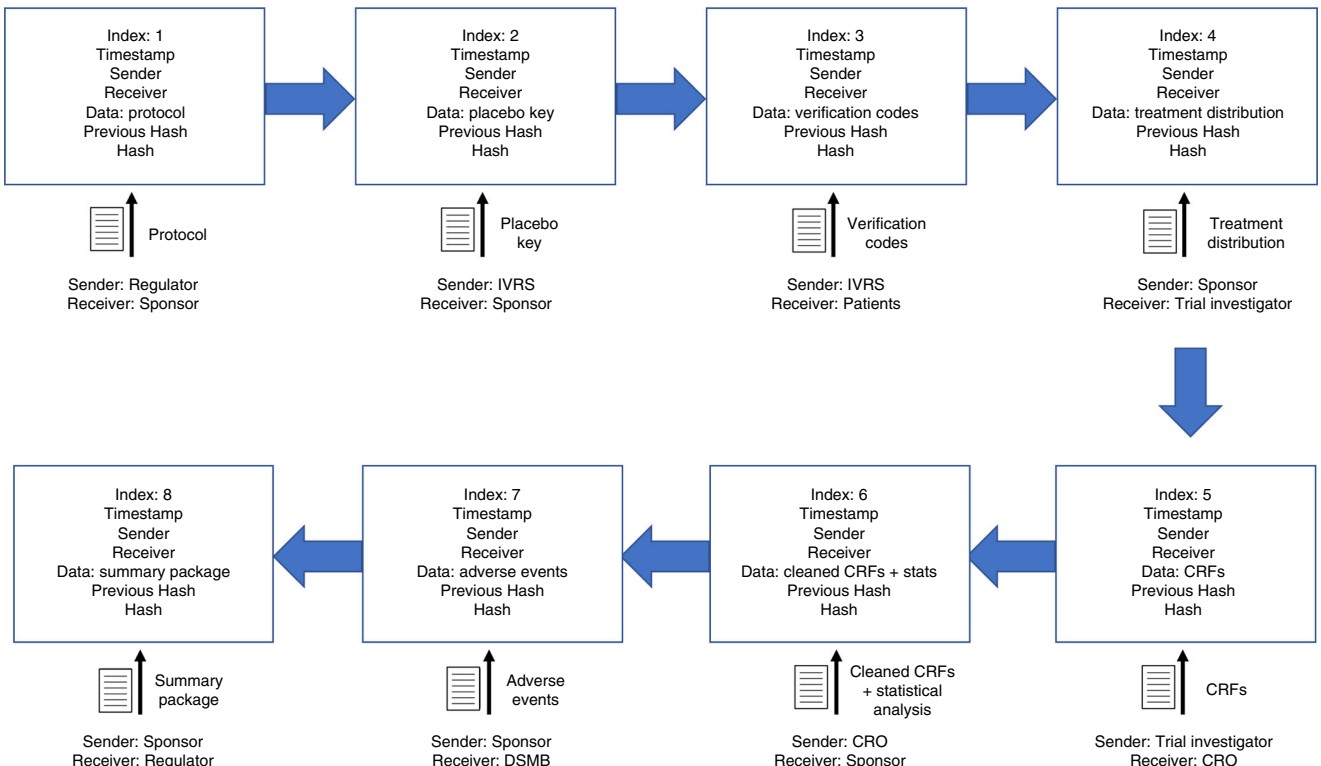

**Fig. 2** The growing blockchain. With each new transaction that occurs, a block is appended and keeps track of information like the timestamp, sender, receiver, file contents, hash of the previous block, and current hash, all in an immutable data structure. On-chain storage of such elements requires minimal memory allocation linearly proportional to the amount of data being uploaded, since the additional book keeping elements are fixed-length strings. Hence scalability is possible, especially given adequate allocation of hardware made possible via growing cloud storage capabilities (see Supplementary Discussion). A summary, compressed blockchain is shown. The actual blockchain will have a beginning genesis block and individual blocks for each transaction (such as a new block for each CRF instead of a single block for all CRFs as shown). The compressed chain is shown to illustrate the chronology of the trial, and what information constitutes a block

on the machine housing the data. Finally, we demonstrate an improved and expedited version of adverse event reporting.

**Patient and clinical investigator encounters**. We composed truncated and digitized CRFs for the four patients we mimicked on the portal using the publicly available CRF component data. For instance, Subject 73,491 from the study came to the clinical site on the first day of the trial period (day 0). The investigator collected immunological data, such as a white blood cell concentration of $5.9 \times 10^3$ cells per μL, eosinophil percentage of 4.6%, and platelet count of $223 \times 10^3$ cells per μL (Supplementary Note 1). In our proposed schema, if the CRF were paper based, then it would be scanned in; if electronically captured, it would be directly added to the growing blockchain via the portal, as is the case in our simulation. Verification codes are appended to each CRF.

**User mediated corruption**. We then simulated two types of hostile conditions for the trial. The first was to simulate an effort to manipulate data that were uploaded from other trial staff. While logged in as the trial sponsor, we attempted to modify the adverse events reported in selected CRFs that recorded subjects 73,491 and 73,511 receiving the treatment drug omalizumab so as to deceptively bolster treatment approval in a potentially untrustworthy network. Subject 73,491 showed many adverse reactions during the treatment period, such as muscle strain, injection site swelling, sinus headaches, and nasal congestion among other events (Supplementary Note 1), while Subject 73,511 exhibited events such as chest tightness, injection site

reactions, sinus congestion, decreased blood pressure, and a lower respiratory tract infection among other ill effects (Supplementary Note 2). As the trial sponsor, these CRFs were mutated such that no adverse events were listed (Supplementary Note 3, 4). The new tampered replacement files are appended with a version number automatically (Fig. 3b), and the corrupting party, time of modification, and changes are all easily visible. The system is capable of handling multiple versions of files in case the original one is part of a later transaction, or in case further revisions or illegitimate mutations are made. These are designated with incrementing version numbers for each new unique version. Original documents, however, are designated with no version number.

With an append only transaction scheme and version controlling, we have a means of keeping a full record of everything that happens to a file, and can easily refer to the author and old and new versions of data, similar to the concept and flexibility of GitHub. This is integral to the auditing process as regulators can track precisely what was changed, by whom, and when with the immutable timestamp. Hence, we simultaneously accommodate the user's need for making changes clear, the regulator's desire for monitoring data easily, and also abide by blockchain's append only schema, which allows for the maintenance and persistence of older data.

**Storage corruption**. The second hostile condition we simulated was that of an intentional fault or data corruption at the storage level. In this simulation, we purposefully corrupted the treatment distribution outlining which medication plan was given to which patients (Supplementary Table 1, 2) to check if the infrastructure

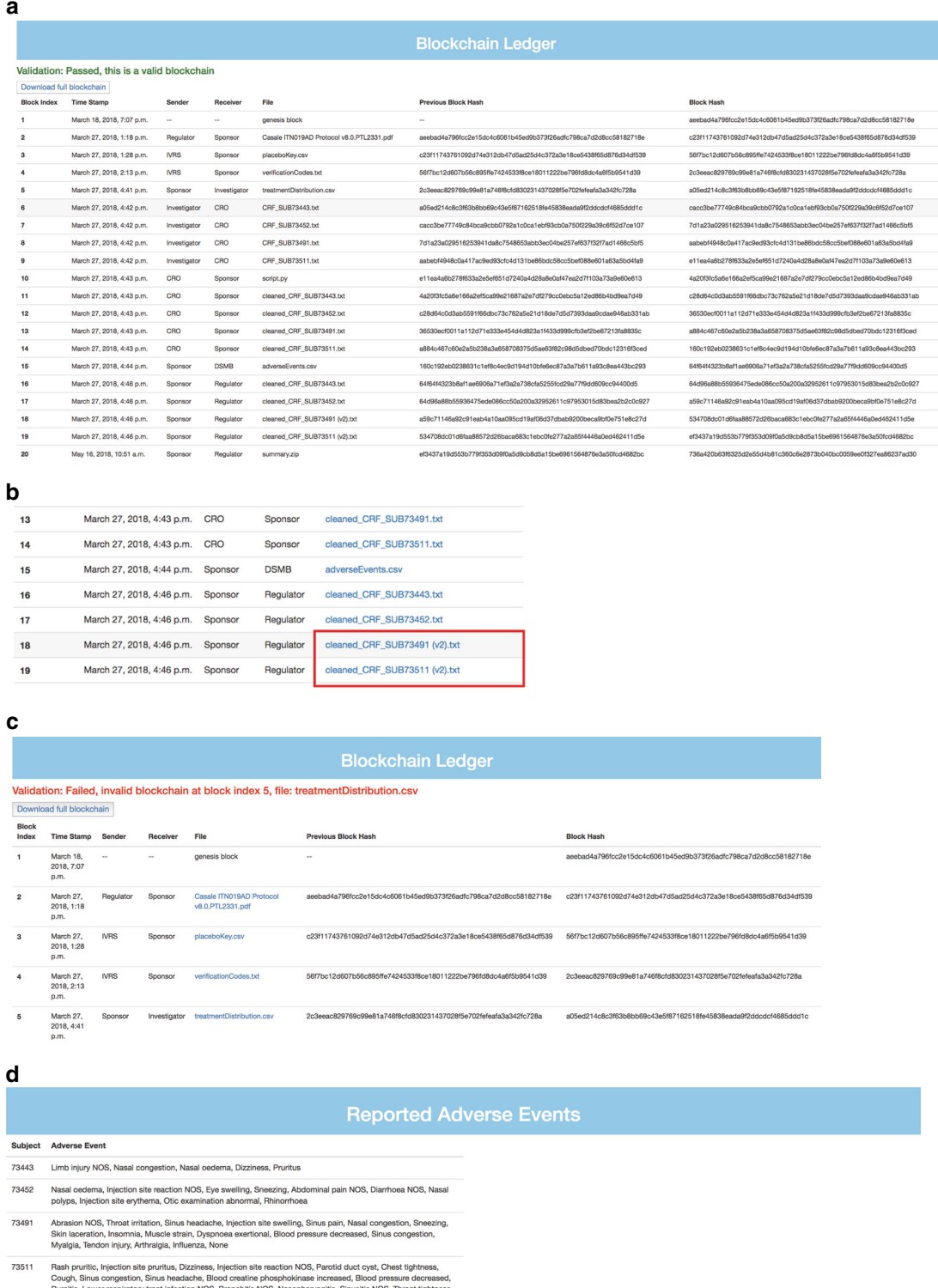

**Fig. 3** Portal functionality. **a** The public ledger shows the full transaction history since the start of the trial. Blocks are timestamped, indexed, and attached with the file of the transaction and identities of the participating parties. The regulator can easily download individual files, or all elements in bulk, and inspect when and between whom files are shared. **b** New versions of files are given a version number, which increments with each new version of that file. In the screenshot above, the original CRF was modified by the sponsor and automatically appended with a (v2) by the system (boxed in red). The responsible party and time of modification are readily apparent. **c** Internal validation and automated hash checks take place to verify the integrity of the data without the need to manually read through the data's contents. Hash checks are performed chronologically starting with the genesis block, and verify that the hash of the contents of the block match what is to be expected (see Supplementary Methods). Validation fails when the treatmentDistribution.csv is modified at the storage level. The precise origin and location of the fault can be readily discerned. **d** Adverse events are auto populated from investigator uploaded CRFs to the pages of the regulator and DSMB, circumventing the normal, slower, and more error prone pathway that is normally taken for adverse event reporting. Such instances are available for inspection at the soonest possible time

would detect and guard against such changes. In the blockchain ledger section of the portal, there is a validation check that successfully shows exactly where the fault and corrupted file lies (Fig. 3c). Due to the sensitivity of a hash function's output in relation to its input, changing the data in even the smallest way in a block, such as modifying a single character in the block's attached file, will result in a completely different hash string. This string will be fed into the input of the next block's hash function, and the resulting string will be completely different from what it was prior to the data modification. Hence, data integrity can be checked by simply comparing the hash strings of a proposed blockchain under audit with a set of verified and correct hashes. Since each transaction is given its own block, precise determination of location and file that were corrupted are possible by simply finding the first block with an incorrect hash. Hence, storage of the desired and correct hashes is necessary, and we advocate for centralized and secured storage by the trusted regulator who will be performing the audit (see Supplementary Methods). Since only hash strings are required for the purpose of verifying integrity, the regulator need not allocate much hard disk space for the audit process. Verifying integrity can be done quickly as the regulator need only check for string equivalence, which is a quick and trivial process.

**Zero-knowledge proof of purity**. In this proof of concept model, we illustrate the ease at which a data repository can be checked and verified for tampering without manually reviewing each file. SHA256 hashing is a quick and highly optimized process[12], and comparing two hash string for equality is trivial. Hence, verifying originality can be done quickly, and without actually opening and inspecting data, which is useful if confidential data are being audited. This serves as a zero-knowledge proof[15] of data integrity because the auditor need not know the exact configurations and detailed information within a file, and yet still verify its originality. This is particularly useful for the data being handled in clinical processes. This type of methodology and proof provides another layer of security and respect for confidential data, all while quickly and automatically verifying purity of data. Furthermore, by giving the user the option of storing encrypted data on the servers, we ensure that sensitive information cannot be compromised even in the event of a data breach.

**Expedited adverse event reporting**. As part of the simulation in the portal, we also scrutinized adverse event reporting and how to improve it. In abidance with the goal of making the management of a clinical trial easier and more effective than the current standard, adverse events from the clinical investigator uploaded CRFs are automatically parsed and populated to the pages of the regulator and DSMB (Fig. 3d). This not only serves as a fast means of assessing safety as the trial continues, but also circumvents the slower and potentially error prone route that adverse event reporting would normally take before reaching the regulator. In the current way clinical trials are run, the CRFs would normally be sent to the CRO, who then parses out the adverse reactions and reports these events to the sponsor, who sends these reports to the DSMB. This process takes time, and is subject to modification or loss by human error or malice. In the proposed scheme and proof of concept service, this vulnerable process is circumvented, and the regulator or DSMB can be immediately notified of each subject's adverse reactions at the soonest possible time, which can be crucial for maintaining public safety. Since the adverse events are extracted directly from the CRFs on the immutable blockchain, the regulator and DSMB can be assured that the events are legitimate.

## Discussion

Blockchain architecture has been prescribed to cure many data interoperability challenges today, including tasks such as overturning the current financial system[6] and remodeling the infrastructure of the internet[16]. Many of these proposed solutions are either not realistic, or have not yet demonstrated efficacy. Here, we showed that a blockchain-based file and data structure could be used to reliably safeguard data in a clinical trials network, and provide an immutable and fully traceable audit trail. Scalability is of utmost importance when considering the utility of such an architecture, especially if it were to move to full-scale production from proof of concept. The system scales linearly in regards to memory allocation. Similarly, performance is also linear in dataset size, so the architecture is able to scale well (for formal discussion, see Supplementary Discussion).

By using the actual clinical trials data from a previously completed major clinical study, Efficacy and Safety Evaluation of Allergen Immunotherapy Co-Administered with Omalizumab, we showed that data entry, storage, and adverse event reporting can be performed in a more robust and secure manner, which could withstand attacks from both other people in the network and infrastructure damage at the storage level.

We do see some limitations in this work. Forcing all participating parties to use a service like this will still remain a challenge. This could be overcome by restricting regulatory approval of a proposed treatment upon the condition that the trial sponsor uses the service for information exchange, and that CROs and trial investigators are used supporting this methodology. By law, many trials that study drugs, biologics, or devices are required to register with the clinicaltrials.gov portal[17], so requiring a registration with a service like this could be feasible.

To encourage use of the service and ensure data integrity, only data on the blockchain would be considered when reviewing potential approval of a drug. Hence, any offline personal transactions of data such as through email would be discouraged. Although blockchain technology provides a means of recording data into structures that are immutable, traceable, and verifiable, it cannot prevent data from being falsified at the point of origin. Clinicians or clinical researchers can be careless or fraudulent and record misleading data into the CRFs, and statisticians within pharmaceutical companies can overinflate $p$-values because of vested interests in success. These mistakes would be carried forward in the blockchain. The ideal would be to encourage the most raw forms of data or input to be captured as early as possible into the blockchain. Regardless, with unfalsifiable data collected from clinical investigators and cleaned by CROs before being sent to the sponsors, regulators would have the raw data to validate the statistical results due to the design of blockchain. Independent statistical analyses can be run to verify the results. The sponsoring statisticians and CROs could also be required to post their Python and R analysis scripts and freeze these on the ledger for verification and reproducibility, though this is not necessary or critical to the blockchain architecture and merely an example of how the chain can be potentially utilized after its creation. More generally, by being able to see each edge in the network and the corresponding transaction and values, regulators are able to confidently cross check the results of any given subsequent transaction.

By providing each patient with a verification code to give to their physician and validate the CRF upon visit, patient interaction is ensured. Note that this is independent to the functionality of the blockchain infrastructure and design, and merely an additional modification we propose to the clinical trials process with feasibility and ease yet to be tested. Unfortunately, if a patient and fraudulent clinician are in full adverse cooperation, there is no way of assuring that real data are generated for that

office visit. In theory, physicians could still fraudulently purchase these verification codes from patients to save time and avoid running the test trial, but adding a verification code provides a deterrent from blatantly fabricating CRFs. Fabricating data will require the involvement and cooperation of the patient. This will be less enticing and riskier for the clinician to generate fake CRFs, as the clinician is no longer the only party with knowledge of the forgery. Discovery of malpractice is now more likely and can make administering the office visit treatment more attractive than fabrication. Additionally, moving clinicians away from paper-based recordings and forms may also be difficult, but not impossible as the move towards electronic health records (EHRs) rises and as the public concern for non-standardized and fallible physical documentation increases. In contrast to physical documentation, which can be convincingly modified, digital documents through our blockchain service cannot be changed without being noticed and invalidated. Accommodating the current standard of paper documentation is still possible with the service, since these can be easily scanned and converted to a digitized data type to be added to the immutable ledger.

We chose to give data storage control to the regulator, as opposed to distributing data storage to nodes across participants in the network like in Bitcoin applications. We preferred this design because the trial regulator is the only party that can and must be trusted, since this party is the one having final approval over success or rejection of the treatment. It is a central authority that cannot be eliminated. We feel implementing distributed storage to everyone in the network (such as that used for Bitcoin) does not fit this regulatory context well, and is also impractical as all parties would also have to locally store data pertinent to the clinical trial on their machines. Requiring usage of a web application is a much more feasible first step, and the regulator can have more refined control over the data and abstract management away from the users.

This service could be an improvement to the current clinical system, and could be integrated into existing workflows with the use of a web-based platform, such as the one prototyped here. A service like this could be especially useful in the context of international clinical trials in which oversight is more difficult to administer[1]. Moreover, there is an added benefit to the regulator as overwatch and monitoring in real-time since the beginning of a trial could be more feasible, and the regulator need not wait until a final summary package is delivered by the sponsor at the end of the trial. However, current regulatory bodies might not be ready to deal with real-time data access.

This framework can also serve to advance the initiative towards open and publicly accessible big data, as the regulator can selectively decide which data elements to release to the public on the portal for each clinical trial. For future directions, the portal can be easily expanded in functionality to include services similar to clinicaltrials.gov, and also open access to raw data for research scientists like the concept behind the ImmPort repository[14]. An interesting future direction and direct application would be to use this methodology to broadcast information that the trial sponsor and regulator deem fit to show to the general public directly through the clinicaltrials.gov portal. This could be an exciting avenue that both takes advantage of the efficiencies in speed and security that this proof of concept has to offer, along with maintaining regulator discretion and control over data. Of course, much work would be needed before this hypothetical direction can be realized. The idea of open access to clinical trials data are a debated topic[18,19], but services like ImmPort have shown that there is a desire among trial sponsors and the scientific community, notably the National Institute of Allergy and Infectious Disease (NIAID) and National Institutes of Health (NIH), to make this data accessible and open[14]. In addition, the International Committee of Medical Journal Editors (ICMJE) has expressed their commitment to promoting clinical research data to be openly available[18].

We propose leaving the question of opening access to trials data to a case-by-case decision of the regulator and sponsor, if they wish to disseminate data on the chain, keeping in mind that the feasibility of maintaining patient privacy and data usability is a question that needs a case-by-case perspective. Our proposal and proof of concept can be synthesized in a methodology that combines ideas of open access data sharing, and the resulting security and trust benefits that arise from blockchain technology. It is important to note however that our blockchain service does not solve or speak to the complex issues of public data sharing, and we are simply advocating this as a potential post-trial use of the service if the regulator deems it responsible and safe. All of the additional functionality that we propose can be integrated into this system, and we will have a unified platform for all things clinical trial catering to the researcher, regulator, and potentially public client. If a service like the one that we propose in this paper can be adopted, the benefits of immutability, traceability, and more trust in the clinical research process can ensue. Regulators can confidently track the complex flow of data throughout a trial, and be kept up to date on safety proceedings and progress.

## Methods

**Portal implementation and simulated trial data**. We built a proof of concept web portal to access a trial-specific blockchain. The portal was built using a Python and Django software development framework, with additional Python encryption libraries, including hashlib and simplecrypt. The portal is hosted on the cloud platform Heroku. A proof of concept prototype of the portal can be found at: http://trialchain.ucsf.edu/. This implementation of the service we are proposing is meant as a proof of concept and is not designed for production usage.

**Code availability**. All source code can be found at https://github.com/wongdaniel8/ClinicalTrials. The code is publicly accessible for view to encourage reproducibility. The software requirements can be found in the GitHub repository, within the requirements.txt file.

**CRF extraction**. In order to test the utility of this service with real world data, we used open access data from a real completed clinical trial: Efficacy and Safety Evaluation of Allergen Immunotherapy Co-Administered with Omalizumab (NCT00078195)[13]. The trial had a total of 159 patients and four treatment arms (see Supplemental Table 2). For the sake of brevity and clarity, we extracted information from four patients in total, one per treatment type. Only a subset of the large wealth of CRF data were mirrored for the simulation. Broad information like race, gender, and age were included. Since the clinical trial was studying allergen immunotherapy, we also extracted relevant assay measurements such as free IgG-a and free IgE-a concentrations in the blood. Additional relevant immunological data, such as concentrations of white blood cells were extracted, and included measurements for monocytes, neutrophils, eosinophils, basophils, and lymphocytes. Concentrations of red blood cells, corpuscular hemoglobin, and platelets were also extracted and recorded into the CRFs. Known adverse events reported during the trial period were also parsed and included in the digitized CRFs. We inserted a necessary verification code on each CRF, with the motivation of encouraging patient and clinician interaction. To meet the requirement of a verification code being present to legitimize the CRF, we simulated the patient providing the verification code (received from the IVRS) to the clinical investigator staff to place on the CRF. Upon audit, the regulator would check that each CRF has the necessary verification code in a process that can be readily machine automated.

## Data availability

We downloaded the completed clinical trials data from the clinical trials data repository ImmPort (SDY1, www.immport.org)[14]. The link to the SDY1 data can be found here: https://www.immport.org/shared/study/SDY1. Elements were downloaded on 2 March 2018.

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

## Acknowledgements

We thank Suman Bhattacharya for his advice on clinical trial data management. This work was supported by the National Institute of Allergy and Infectious Diseases (Bioinformatics Support Contract HHSN316201200036W). The content is solely the responsibility of the authors and does not necessarily represent the official views of the National Institutes of Health.

## Author contributions

D.R.W. built the prototype and wrote and edited the manuscript. S.B. helped with ascertaining the clinical network topology, retrieving ImmPort data, editing, and supervision. A.B. aided in the writing and editing process, and provided ample supervision and guidance in the formulation of the application.

## Additional information

**Competing interests:** The authors declare no competing interests.

