## [Peer Review File · Nature Communications]

Reviewers' comments:

Reviewer #2 (Remarks to the Author):

Thank you for the opportunity to review the submitted work, "Prototype of Running Clinical Trials in an Untrustworthy Environment Using Blockchain". The authors seek to address many of the existing shortcomings with current clinical trial processes by means of a relatively new platform and data structuring approach. The intent of this PoC looks to map the strengths of blockchain technology to help address deficits (e.g., transparency, accessibility, security) in the current systems used for clinical trials. Notably, the described work would represent a substantial advancement as it extends previous published efforts (e.g., Nugent) from proposed to an actual build. Please find suggestions and comments below corresponding to manuscript line, followed by comments of a broader nature.

1 Given the non-distributed nature of storage described and the requisite trust required of the regulator, title still does not conflict with these elements due to phrasing employed.

Supp 34/349 Would be interesting to see this measured. This is outside of the scope of the current submission.

54-56 Would consider adding complementary work on enforcing human subject regulations using blockchain and smart contracts (<https://doi.org/10.30953/bhty.v1.10>) to more broadly outline efforts in this arena.

54-56 Would consider adding exploration of healthcare and geospatial blockchain (<https://doi.org/10.1186/s12942-018-0144-x>) to further demonstrate breadth of blockchain in healthcare beyond settings currently cited.

122 Figure 2. Logically presented process and abbreviations in processes are addressed nicely in description.

129 Would consider briefly addressing conventional concerns about on-chain storage of health-related data (size/exposure).

137 Version control is a rational approach to balance the need for editing and desire for immutability in this environ.

275 Interesting development as this type of implementation could have a secondary public health benefit...albeit difficult to measure.

281-310 Figure 3 leverages trialchain and sources nicely to help illustrate relevant points. 3A&3D may be particularly useful for visualizing output to readers less familiar with the underlying tech.

331 Would you foresee an integrated system where outputs from this framework could be automated to populate clinicaltrials.gov or a next gen version, to both satisfy regs and build upon the gained efficiencies in speed and compliance that this PoC appears to do (as suggested in 386)?

327 Recommend addressing potential challenges in scaling (from data subset of one study to a national system) that have been seen as other testnets transition to mainnets (e.g., congestion, loss of velocity, etc.). While those are primarily permissionless constructs, it still bears mention and would reflect an appropriate proactive measure here.

Recognizing that this may be intended as a brief communication and that (modified) blockchain protocol and consensus selection/generation may be purposefully ambiguous, it is more difficult to

assess some aspects of the described PoC beyond a conceptual nature. This is unfortunate as it is those pragmatic elements that really distinguish the current work as a potentially meaningful advance beyond the theoretical that exists in the literature. **In lieu of the preceding, it may be easier to create a more explicit/earlier callout in the text or Supplemental Methods for the (updated) readme in Github? Similarly, would like to have seen a little more detail on architecting the regulator component as it is such a core component; however, this level of detail may not be possible in this submission type.

Minor edits:

Spell out first time use of abbreviations (e.g., FDA, API, etc.)

Citation sequencing needs to be reviewed. For example, it appears References 3 and 4 are reversed so that the text throughout intends to refer to #3 for the Nakamoto white paper and #4 for the Kuo health care uses.

Reviewer #3 (Remarks to the Author):

This paper argues that blockchain technology has the potential to improve the clinical trial process. The claim appears reasonable at face value, and certainly addresses an acute need to make clinical research more efficient as well as cost-effective. The authors have performed a limited proof of concept of conducting a trial using blockchain technology. Although the paper raises several interesting issues, it does not adequately address them, and is too superficial to be fully convincing.

The statement that “there is a lack of transparency and traceability of data” in clinical trials appears too abrupt to be taken at face value. An industry of electronic data capture has blossomed in clinical research, and whilst it is certainly true that the proprietary systems in place are both inefficient and expensive, they offer full traceability of data transactions as well as effective protections against data tampering.

There are obvious advantages to blockchain technology, primarily the ease of maintaining and checking the integrity of data, but also the speed with which information can be shared among all parties that might need to review it. The example of expedited safety reporting is excellent and would be a huge improvement over the awkward and unreliable procedures in place for such reporting in the vast majority of trials conducted in the US and elsewhere.

The authors mention data fabrication as a key reason to use blockchain technology. This has been an area of intense research, and a single reference to a survey of investigators reporting to be “personally aware of intentional fabrication in research” does not do justice to the topic. Moreover, it is unclear that block chain technology will be able to prevent a substantial amount of data fabrication, which come before the data are submitted to a central database in multicentre clinical trials.

The authors’ proposal that “the Food and Drug Administration in the United States instantiates a private blockchain and registers all participating parties” is not realistic, because the FDA is not (and should not be) mandated to get involved in the actual conduct of clinical trials. It would therefore not be appropriate for them to have control over a blockchain-driven clinical trial program, though they could be passive users of the information, e.g. for safety review purposes. The authors claim that distributed data storage as in Bitcoin applications would not fit the need of clinical research applications, yet they provide no arguments for their position. Even if they convincingly argued that one central party needs to have control over a clinical trial, then why could it not be the sponsor of the trial, or a CRO appointed by the sponsor?

Many of the authors' ideas are commendable, but could be implemented outside of blockchain technology. For instance, the authors ask that "The pharmaceutical statisticians and CROs can be required to post their Python and R analysis scripts and freeze these on the ledger for verification and reproducibility". This proposal is consistent with the principles of reproducible research which should be (but are not) imposed on clinical research, whether blockchain technology is used or not.

The authors propose to provide "each patient with a verification code to give to their physician and validate the CRF upon visit". Such a patient involvement could potentially add much value to patient-centric clinical trials, but again such an improvement to the clinical trial process seems largely independent of the use of blockchain technology, and might not be as easy to implement as the authors portray.

Finally, the authors argue that blockchain technology would facilitate "open access to raw data for research scientists". While the present reviewer wholeheartedly agree with them that such access would be extremely valuable and would maximize potential uses of precious clinical trial data, the issue of open access to clinical trial data has been passionately debated in recent years. The authors do not refer to any of the numerous and thoughtful publications on this topic. The authors only briefly touch on extremely complex issues, and it is unclear that blockchain technology would help address some of the difficult problems of open data access – including patient privacy and interpretability of the data, among others.

Response to Referees

We would first like to thank all of the referees who took the time to thoroughly read our manuscript and provide us with helpful feedback and recommendations to improve our manuscript. The reviews included valuable and diverse insight that we had not considered, and we thank the referees for bringing these to our attention. We have considered all of the suggestions for improvement, and made corresponding changes in the manuscript. Listed below is a point by point address for each comment we received. The comments are included for reference in blue text, and our address in black. Thank you for your consideration.

Reviewer #2 (Remarks to the Author):

Thank you for the opportunity to review the submitted work, “Prototype of Running Clinical Trials in an Untrustworthy Environment Using Blockchain”. The authors seek to address many of the existing shortcomings with current clinical trial processes by means of a relatively new platform and data structuring approach. The intent of this PoC looks to map the strengths of blockchain technology to help address deficits (e.g., transparency, accessibility, security) in the current systems used for clinical trials. Notably, the described work would represent a substantial advancement as it extends previous published efforts (e.g., Nugent) from proposed to an actual build. Please find suggestions and comments below corresponding to manuscript line, followed by comments of a broader nature.

1 Given the non-distributed nature of storage described and the requisite trust required of the regulator, title still does not conflict with these elements due to phrasing employed.

Noted. We were careful with title wording, and the non-distributed nature of storage is described in supplemental.

Supp 34/349 Would be interesting to see this measured. This is outside of the scope of the current submission.

Agreed, the idea of patient verification codes is not the main focus of our study and discussed as an additional way in which we propose to change the clinical trial process, for which this does not necessitate the use of blockchain. We added a comment on line 437 about its limitation and how we have not tested or measured this yet.

54-56 Would consider adding complementary work on enforcing human subject regulations using blockchain and smart contracts (<https://doi.org/10.30953/bhty.v1.10>) to more broadly outline efforts in this arena.

Noted and added on line 99

54-56 Would consider adding exploration of healthcare and geospatial blockchain (<https://doi.org/10.1186/s12942-018-0144-x>) to further demonstrate breadth of blockchain in healthcare beyond settings currently cited.

Agreed, the field of blockchain technology and its applications to healthcare and medicine are exploding with new information. We have added the recommended review to line 95 and expanded our breadth and survey of blockchain's potential applications.

122 Figure 2. Logically presented process and abbreviations in processes are addressed nicely in description.

Noted.

129 Would consider briefly addressing conventional concerns about on-chain storage of health-related data (size/exposure).

Agreed, potential storage concerns are addressed on line 183 and points to an in-depth explanation in the supplemental methods. In regards to potential exposure concern, we added a short comment on line 169 about integrity and limiting unwanted exposure through cryptography.

137 Version control is a rational approach to balance the need for editing and desire for immutability in this environ.

Agreed.

275 Interesting development as this type of implementation could have a secondary public health benefit...albeit difficult to measure.

Agreed, we believe this can potentially provide additional adherence to safety and awareness of events.

281-310 Figure 3 leverages trialchain and sources nicely to help illustrate relevant points. 3A&3D may be particularly useful for visualizing output to readers less familiar with the underlying tech.

Agreed, the interface is point and click and requires no coding or advanced technological experience to use.

331 Would you foresee an integrated system where outputs from this framework could be automated to populate clinicaltrials.gov or a next gen version, to both satisfy regs and build upon the gained efficiencies in speed and compliance that this PoC appears to do (as suggested in 386)?

Indeed, we do foresee this possibility and are hopeful that we can put this service to greater action, with clinicaltrials.gov being an obvious outlet. We added a brief commentary on line 485 expressing this potential avenue, and more discussion on open sharing of data on line 508.

327 Recommend addressing potential challenges in scaling (from data subset of one study to a national system) that have been seen as other testnets transition to mainnets (e.g., congestion,

loss of velocity, etc.). While those are primarily permissionless constructs, it still bears mention and would reflect an appropriate proactive measure here.

Agreed, the previous manuscript was lacking in discussing scalability. We have added a detailed discussion on scalability in the supplemental methods with a new section entitled “Scalability”, which is pointed to in the main text (on lines 186 and 389) if the reader wants to explore this.

Recognizing that this may be intended as a brief communication and that (modified) blockchain protocol and consensus selection/generation may be purposefully ambiguous, it is more difficult to assess some aspects of the described PoC beyond a conceptual nature. This is unfortunate as it is those pragmatic elements that really distinguish the current work as a potentially meaningful advance beyond the theoretical that exists in the literature. **In lieu of the preceding, it may be easier to create a more explicit/earlier callout in the text or Supplemental Methods for the (updated) readme in Github? Similarly, would like to have seen a little more detail on architecting the regulator component as it is such a core component; however, this level of detail may not be possible in this submission type.

Agreed some points of the manuscript are difficult to assess and test, and outside of the scope of our aims and claimed improvements. When these occurrences happen, we are sure to state the limitation. We do however offer improvements to the current system and test these improvements. The GitHub README has been updated to inform the reader of the conceptual proof of concept nature, and how consensus is mirrored programmatically as opposed to through hardware modifications that would be necessary if the proof of concept is to be fully productionized and made large scale. To advocate for potential pragmatism, we added a discussion on scalability and what requirements would be needed to productionize the proof of concept in the new “Scalability” section of the Supplement. We show that both memory and performance is bounded linearly, and hence we suggest feasibility and practicality. Details regarding architecting the regulator component was minimal in the last manuscript, and we’ve added more details in the supplemental under the new supplemental section “Permissioning” if the reader is curious. Luckily this process of setting up a regulator is minimal effort and very similar to instantiating all other nodes. We believe that these details are supplemental, and agree that a full explanation of how the software exactly executes this is not appropriate for this submission type (although this information is readily available in the GitHub repository).

Minor edits:

Spell out first time use of abbreviations (e.g., FDA, API, etc.)

Corrected.

Citation sequencing needs to be reviewed. For example, it appears References 3 and 4 are reversed so that the text throughout intends to refer to #3 for the Nakamoto white paper and #4 for the Kuo health care uses.

Corrected.

Reviewer #3 (Remarks to the Author):

This paper argues that blockchain technology has the potential to improve the clinical trial process. The claim appears reasonable at face value, and certainly addresses an acute need to make clinical research more efficient as well as cost-effective. The authors have performed a limited proof of concept of conducting a trial using blockchain technology. Although the paper raises several interesting issues, it does not adequately address them, and is too superficial to be fully convincing.

The statement that “there is a lack of transparency and traceability of data” in clinical trials appears too abrupt to be taken at face value. An industry of electronic data capture has blossomed in clinical research, and whilst it is certainly true that the proprietary systems in place are both inefficient and expensive, they offer full traceability of data transactions as well as effective protections against data tampering.

We have “softened” these claims in the introduction, as suggested. The previous manuscript did not cite enough evidence for the claim, “there is a lack of transparency and traceability of data”. Hence, we’ve added an additional two peer reviewed sources to back our claim about lack of transparency and traceability on line 57 and 58. The current services are indeed inefficient and inexpensive, but traceability is still limited and our proof of concept addresses some of these issues.

There are obvious advantages to blockchain technology, primarily the ease of maintaining and checking the integrity of data, but also the speed with which information can be shared among all parties that might need to review it. The example of expedited safety reporting is excellent and would be a huge improvement over the awkward and unreliable procedures in place for such reporting in the vast majority of trials conducted in the US and elsewhere.

Agreed, the current method is slow and time is wasted. The current clinical trials methodology does not really facilitate real-time access to data, and is notoriously inefficient.

The authors mention data fabrication as a key reason to use blockchain technology. This has been an area of intense research, and a single reference to a survey of investigators reporting to be “personally aware of intentional fabrication in research” does not do justice to the topic. Moreover, it is unclear that block chain technology will be able to prevent a substantial amount of data fabrication, which come before the data are submitted to a central database in multicentre clinical trials.

Noted, we have added a citation to a review of the topic on line 47 to add more breadth and awareness to the topic, and have “softened” the wording around this claim as well. It is unclear how much data fabrication will be prevented (this is innately a difficult measurement to test) but we do offer some protection as demonstrated in the Results section. Detection of fraud in this case guards against one node corrupting data set forth by another node, which may be of use in multi-site clinical trials in which there are many parties with many agendas. There is no way to prevent a node’s fabrication of its own new data which the network has not yet seen, and this limitation and inability to guard original data was stated in line 407.

The authors' proposal that "the Food and Drug Administration in the United States instantiates a private blockchain and registers all participating parties" is not realistic, because the FDA is not (and should not be) mandated to get involved in the actual conduct of clinical trials. It would therefore not be appropriate for them to have control over a blockchain-driven clinical trial program, though they could be passive users of the information, e.g. for safety review purposes. The authors claim that distributed data storage as in Bitcoin applications would not fit the need of clinical research applications, yet they provide no arguments for their position. Even if they convincingly argued that one central party needs to have control over a clinical trial, then why could it not be the sponsor of the trial, or a CRO appointed by the sponsor?

Noted, we have reworded that statement on line 108 so as not to necessitate the FDA as the regulator and to make the identity of the regulator more open. We defended the claim that data storage as in bitcoin applications would not be fitting in lines 462 – 470 and also in line 88. We addressed the potential issue of having the sponsor also control the trial and chain in the Results section under "User Mediated Corruption", in which the sponsor corrupts data to bolster the apparent safety of the drug. The sponsor has a vested interest in success of the trial, so perhaps this would not be the best entity to also fulfill the regulator position. The appropriate government regulatory agency with no financial incentive would likely be a better option. The FDA, who already currently regulates clinical trials and is the one necessary trusted entity in the network in the United States, can be a potential candidate to fulfill this role, but we do not mandate this.

Many of the authors' ideas are commendable, but could be implemented outside of blockchain technology. For instance, the authors ask that "The pharmaceutical statisticians and CROs can be required to post their Python and R analysis scripts and freeze these on the ledger for verification and reproducibility". This proposal is consistent with the principles of reproducible research which should be (but are not) imposed on clinical research, whether blockchain technology is used or not.

Noted, we have made sure to specify that this is not necessary to the blockchain architecture, but merely an example use of the architecture. The more general use is stipulated on line 432 and is specific to the blockchain ledger.

The authors propose to provide "each patient with a verification code to give to their physician and validate the CRF upon visit". Such a patient involvement could potentially add much value to patient-centric clinical trials, but again such an improvement to the clinical trial process seems largely independent of the use of blockchain technology, and might not be as easy to implement as the authors portray.

Agreed, we have made sure to add a note on line 437 that this is an additional modification we are proposing independent of blockchain, with feasibility yet to be tested. We are sure to make no claim about this being easy to do.

Finally, the authors argue that blockchain technology would facilitate "open access to raw data for research scientists". While the present reviewer wholeheartedly agree with them that such access would be extremely valuable and would maximize potential uses of precious clinical trial data, the issue of open access to clinical trial data has been passionately debated in recent years.

The authors do not refer to any of the numerous and thoughtful publications on this topic. The authors only briefly touch on extremely complex issues, and it is unclear that blockchain technology would help address some of the difficult problems of open data access – including patient privacy and interpretability of the data, among others.

Agreed, we have added more discussion on this topic and a citation to a paper discussing the merits and objections of open data sharing on line 509, and an article advocating for more open access to clinical data by International Committee of Medical Journal Editors on line 513. We have also made sure to state that our blockchain application does not speak to solving any of these complex issues, but rather open data sharing is a potential use case of the chain on line 520.

REVIEWERS' COMMENTS:

Reviewer #2 (Remarks to the Author):

All reviewer comments have been satisfactorily addressed in this revision.

Reviewer #3 (Remarks to the Author):

The authors have adequately addressed the reviewers' comments. They should carefully check their list of references. Only the first author is mentioned, which is not the usual convention.

Moreover, one additional reference is incorrect:

Stephen, G. Data fraud in clinical trials.

should be

George, SL; Buyse, M. Data fraud in clinical trials.

Response to Referees

Reviewer 2 had no further edits or suggestions for improvement:

“All reviewer comments have been satisfactorily addressed in this revision.”

Reviewer 3 had the following to say:

“The authors have adequately addressed the reviewers' comments. They should carefully check their list of references. Only the first author is mentioned, which is not the usual convention.

Moreover, one additional reference is incorrect:

Stephen, G. Data fraud in clinical trials.

should be

George, SL; Buyse, M. Data fraud in clinical trials.”

This has been corrected and we thank reviewer 3 for the attention to detail.